



# Increased flow rate of hyperpolarized aqueous solution for dynamic nuclear polarization-enhanced magnetic resonance imaging achieved by an open Fabry–Pérot type microwave resonator

**Alexey Fedotov[1], Ilya Kurakin[1], Sebastian Fischer[2], Thomas Vogl[2], Thomas F. Prisner[3], and Vasyl Denysenkov[3]**

[1]Institute of Applied Physics of the Russian Academy of Sciences, Nizhny Novgorod, 603950, Russia
[2]Institute of Diagnostic and Interventional Radiology, University Hospital Frankfurt,
Frankfurt am Main 60590, Germany
[3]Institute of Physical and Theoretical Chemistry and Center of Biomolecular Magnetic Resonance,
Goethe University, Frankfurt am Main 60438, Germany

**Correspondence:** Thomas F. Prisner (prisner@chemie.uni-frankfurt.de)
and Vasyl Denysenkov (denysenkov@em.uni-frankfurt.de)

**Abstract.** TS1 A continuous flow dynamic nuclear polarization (DNP) employing the Overhauser effect at ambient temperatures can be used among other methods to increase sensitivity of magnetic resonance imaging (MRI). The hyperpolarized state of water protons can be achieved by flowing aqueous liquid through a microwave resonator placed directly in the bore of a 1.5 T MRI magnet. Here we describe a new open Fabry–Pérot resonator as DNP polarizer, which exhibits a larger microwave exposure volume for the flowing liquid in comparison with a cylindrical $TE_{013}$ microwave cavity. The Fabry–Pérot resonator geometry was designed using quasi-optical theory and simulated by CST software. Performance of the new polarizer was tested by MRI DNP experiments on a TEMPOL aqueous solution using a blood-vessel phantom. The Fabry–Pérot resonator revealed a 2-fold larger DNP enhancement with a 4-fold increased flow rate compared to the cylindrical microwave resonator. This increased yield of hyperpolarized liquid allows MRI applications on larger target objects.

## 1 Introduction

Magnetic resonance imaging is a widespread method in clinical diagnostics providing in vivo information on physiological and metabolic changes in tissue. MRI is a non-invasive method because of the low excitation frequency in the radiofrequency range. However, this implies also the low sensitivity of the method, leading to long acquisition times. This can be improved by moving to higher magnetic fields, but on the cost of high prices and large technical demands. Contrast between different tissues is another concern. Contrast agents, as for example Gd-complexes are typically used to improve the contrast. However, in few cases, it induces side effects like allergic reactions and nephrogenic systemic fibrosis (Perazella, 2009; Thomsen, 2008; Kuo et al., 2007). Besides, potential long-term implications arising from the permanent deposition of the Gd-complexes in the brain are currently under discussion (Semelka et al., 2016; Topcuoglu et al., 2020). As an alternative, hyperpolarization methods have been explored to improve sensitivity and contrast for MRI applications. Different strategies have been used so far to hyperpolarize an agent for MRI applications. Optically polarized noble gases as well as para-hydrogen have been used as hyperpolarized molecules for MRI (Salerno et al., 2001; Albert and Balamore, 1998; Limburn et al., 2013; Golman et al., 2001; Duckett and Mewis, 2012; Cavallari et al., 2018). An-

other method is DNP (dynamic nuclear polarization) where unpaired electron spins excited by microwaves are used to hyperpolarize nuclear spins. Initially this method has been used either at very low magnetic fields or with rapid field cycling (Foster et al., 1998) because microwave radiation cannot penetrate deep into living tissue, or later by hyperpolarizing molecules externally before injection. Nitroxide radicals like TEMPOL are widespread in use for MRI DNP. They have a high Overhauser DNP transfer efficiency to water protons, especially at higher magnetic field strengths (Krummenacker et al., 2012). Furthermore, these radicals can be easily scavenged after the polarization or dissolution step which makes samples biologically compatible and also reduces the shortening of the nuclear $T_1$ relaxation time by the radical (Mieville et al., 2010). However, EPR spectra of nitroxides have three $^{14}N$ hyperfine lines that hampers a complete saturation of the EPR spectrum at lower radical concentrations. Additionally, for potential medical MRI applications the high radical concentrations needed for Overhauser DNP are not suitable for injection. They can be avoided by immobilization of the nitroxide radicals inside the microwave resonance structure on beads (McCarney and Han, 2008), but the altered dynamics makes the DNP experiments somewhat more complicated and less efficient.

Dissolution DNP is a commercially available hyperpolarization method of liquid substrates for MRI (Olsson et al., 2006). It uses metabolites such as $^{13}C$-pyruvate, which are polarized externally to the MRI magnet at very low temperatures ($\sim 1\,K$) by the solid effect. Within 5 min a polarization of 10 % can be achieved (Bornet and Jannin, 2016). After the hyperpolarization procedure is complete, the frozen substrate is rapidly transported to the imager magnet, dissolved in hot liquid, and injected into the body. The typical (dissolving and transfer) time of 5–15 s limits the in vivo usage to $^{13}C$ and $^{15}N$. It is difficult to detect hyperpolarized protons with this approach because of the short relaxation time of water protons compared to the transfer time (Ardenkjaer-Larsen et al., 2014). Therefore, the method is mostly used for the study of cellular metabolism with pyruvate, lactate or other metabolic precursors labelled with low-gyromagnetic ratio nuclei having long relaxation times (Mishkovsky et al., 2012; Jannin et al., 2019). The observation time window is limited by the relaxation time of the hyperpolarized nuclei. Long-lived singlet states might allow to extend this time window (Pileio et al., 2006; Dumez et al., 2017).

Hyperpolarization by the Overhauser mechanism is well suited for physiological solutions under continuous flow conditions, because of the rapid polarization transfer from the unpaired electrons of the radicals to proton spins of the solvent. The polarization transfer in liquids works most efficiently at low magnetic fields. Therefore, first Overhauser DNP experiments for MRI (Guiberteau and Grucker, 1997; Krishna et al., 2002), and proton-electron double-resonance imaging (PEDRI) (Foster et al., 1998) were demonstrated at low magnetic fields of 6–15 mT. This allows irradiation of the paramagnetic radical inside the mouse or rat body in the MRI magnet. Such experiments, however, cannot be performed in clinical MRI scanners at higher magnetic fields (respectively at higher frequencies) due to the strong absorption of microwaves in living tissues, resulting in unwanted heating. A second approach is again to hyperpolarize molecules outside of the target object. This can be done by placing a mw resonator with the radicals into the fringe field of the MRI magnet (Lingwood et al., 2012) or directly inside the MRI magnet close to the target object (Krummenacker et al., 2012). In the first case the hyperpolarization step is done at a lower magnetic field compared to the MRI detection field. This gives a higher polarization transfer efficiency, because the Overhauser effect in liquids is optimal at low magnetic fields. However, the maximal enhancement achieved at the imaging object is downscaled by the ratio between the polarizing and the imaging detection field. For example, in case of an X-band polarizer operating at $B_0 = 0.3\,T$ and a 1.5 T MRI scanner this reduction factor is 5-fold. In both cases the fast polarization build-up times allow for continuous delivery of hyperpolarized water, which is essential for monitoring flow and perfusion. In addition, if the hyperpolarization of water reaches a steady state, it eliminates the need for fast MR imaging sequences. In vivo injection of hyperpolarized water safely allows for perfusion imaging in interstitial spaces, localized angiography, and the visualization of brain perfusion because hyperpolarized water freely crosses the blood-brain barrier (Lingwood et al., 2012). Limitations in the accessible area arises from the fast proton spin relaxation.

In our "in-bore" 1.5 T MRI DNP setup there is no such "Boltzmann factor penalty" and a water proton signal enhancement of up to 100 has been reached under static conditions (Krummenacker et al., 2012). On the other hand, the dimensions of fundamental resonators scale down inversely with the mw frequency. Technical challenges are due to the unavoidable high dielectric losses of water in the mw frequency range (Neumann, 1985). These dielectric losses of aqueous samples result in a decrease of the resonators $Q$-factor, and strong heating of the liquid. The heating can be minimized by applying a resonance mw structure with well separated $E$ and $B$ components of the applied mw field and by placing the sample in a node of the $E$ component. Therefore, cross-section dimensions of the liquid sample inside the resonator have to be much smaller than the mw wavelength. This leads also to very short retention times in case of the flowing liquid. The attainable yield is defined by the highest possible flow rate at a maximum DNP enhancement in such non-stop-flow polarizers. It depends on: (1) the dwell time of water molecules inside the mw polarizer, which should be longer than the proton relaxation time for an optimal DNP build-up, and (2) the mw $B_1$ field strength, which defines the efficiency of saturation of the EPR transitions of the dissolved TEMPOL radicals. The $B_1$ value at the sample depends on the conversion factor $c$ and the quality factor $Q$ of the mw resonance structure, and the applied mw power $P_{mw}$

(Eq. 1) (Poole, 1967; Rinard and Eaton, 2005).

$$B_1 = c\sqrt{Q \cdot P_{\mathrm{mw}}} \qquad (1)$$

The quality factor $Q$ of the resonator itself depends on the geometry of the mw resonance structure and on sample properties, such as size of the sample and its dielectric constant $\varepsilon_r$.

Conveniently, cylindrical capillaries made of quartz can be used as liquid sample holders. For this reason, the natural choice of the mw resonator geometry is also cylindrical geometry, such as $TE_{01n}$ type cavities. Such resonators have high $Q$ factors as well as high conversion factors $c$ for frozen solution samples and are commonly used in EPR spectroscopy (for instance, $Q = 4000$ for an unloaded $TE_{012}$ resonator at 95 GHz mw frequency has been reported (Smirnov and Smirnova, 2001). This leads to a large $B_1$ field strength, which is advantageous for pulsed EPR applications. The situation is quite different for MRI DNP applications. Here the aqueous sample is in the liquid state, strongly increasing the mw absorption if the sample capillary diameter is not strongly reduced. On the other hand, larger amounts of hyperpolarized liquids are preferable to be able to enhance contrast of as large as possible objects under MRI study.

Fabry–Pérot resonators are mw resonance structures suitable for larger sample volumes, because different from fundamental mode mw cavities, the sample size is only limited in one dimension (direction of the mw excitation) much less than wavelength to avoid absorption of the $E$ field component (Budil and Earle, 2004). Among known quasi-optical mw resonators (Lynch et al., 1988; Prisner et al., 1992; Barnes and Freed, 1997; Milikisiyants et al., 2018; Blok et al., 2004; van Tol et al., 2005; Neugebauer and Barra, 2010) a semi-confocal Fabry–Pérot resonator with the sample placed on the surface of a massive metal (copper) block acting as the plain mirror can best satisfy heat dissipation conditions (Denysenkov and Prisner, 2012).

Here we report on such a new open resonator for Overhauser DNP applications within a whole-body 1.5 T MRI scanner. Particular effort was directed at reaching a uniform (plateau like) microwave field distribution along the flow direction to obtain an efficient polarization transfer to water molecules in the polarizer. The performance of the "in-bore" MRI DNP polarizer equipped with the new resonator was tested and compared to the cylindrical $TE_{013}$ cavity described before (Denysenkov et al., 2017).

## 2 Methods and materials

### 2.1 Design and simulation of the resonator

An increase of the flow rate for a constant residence time of water molecules inside the mw cavity requires a corresponding increase of the cavity dimensions in flow direction. This can result in a higher spectral density of mw modes of the

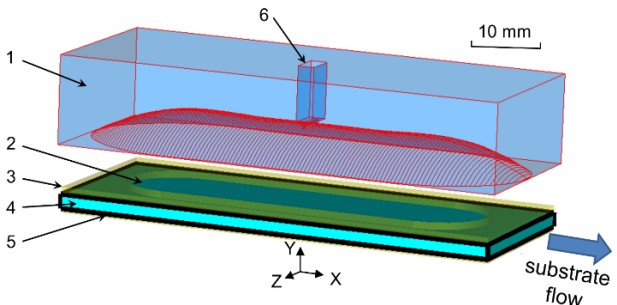

**Figure 1.** Design of open mw resonance structure for MRI DNP. 1 – concave mirror (copper), 2 – aqueous sample, 3 – fixture (brass), 4 – cover plate (sapphire), 5 – plain mirror (copper), 6 – waveguide port. Aqueous solution flow direction is indicated by the "substrate flow" arrow.

cavity, which could prevent the selective excitation of the desired mode. Excitation of other modes with similar frequency would distort the standing mw field pattern inside the cavity and thus reduce the excitation efficiency. In contrast, the use of a Fabry–Pérot open resonance structure avoids such problems. Moreover, the semi-confocal arrangement, where the thin layer liquid sample is directly placed on top of the flat copper mirror, allows effective cooling. A schematic drawing of the new Fabry–Pérot resonator for DNP is depicted in Fig. 1.

Typically, Fabry–Pérot resonators use concave mirrors of elliptical or parabolic geometry. These mirrors provide Gaussian (or quasi-Gaussian) transverse distribution of the microwave field intensity at the mirror surface. However, for the considered DNP application a uniform distribution of the $B_1$ field along the flow direction is preferable. It facilitates the polarization transfer from the unpaired electron spin of the TEMPOL radical to the proton spin of the water molecules, within their residence time inside the resonance structure. Therefore, the shape of the concave top mirror was calculated using an inverse design procedure, as described before (Belousov et al., 2000). In this approach, the $B_1$ field distribution at the flat mirror is assumed as quasi-homogeneous with the same phase. With this boundary condition, the field propagation inside the resonator is calculated.

The concave mirror shape is found as a surface where the phase of the calculated field is equal to $q\pi$, were $q$ is an integer (in our case, $q = 4$, so that we use the 4th axial mode of the cavity). In these simulations the exact equations of the microwave field propagation are used instead of the paraxial approximation for higher accuracy. The possible influence of the finite sizes of both the flat and the concave mirrors was analyzed by an iterative method after the mirror shape was fixed (Fox and Li, 1961). For this purpose, the microwave field propagation from one mirror to the other one was iteratively calculated. For each passage, the phase of the microwave field components was inverted at the mirror sur-

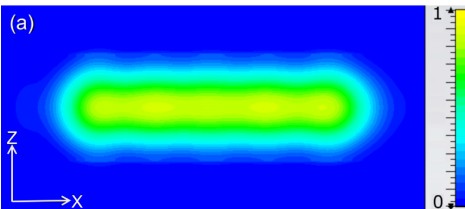 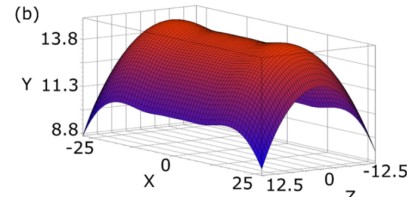

**Figure 2.** Microwave magnetic field distribution in open resonator. **(a)** The simulated microwave magnetic field distribution (normalized) at the plain mirror. The substrate flow is along the horizontal $x$ direction. **(b)** The calculated shape of the concave mirror (scale is in mm).

faces, while the mw power traveling outside of the mirror surfaces was assumed as the power lost. This procedure provides easily the first eigenmode of the resonator with the highest $Q$ factor. The eigenfrequency of this mode is found by satisfying the standing wave condition that the phase change along the wave round-trip in the cavity is equal to $2\pi q$. The quality factor $Q$ of the empty resonance structure can be determined by the mw losses per cycle due to finite mirror sizes. The simulations show that the influence of the finite mirror size are much less than the power loss arising from the liquid sample and therefore negligible. The obtained $B_1$ field distribution at the surface of the flat mirror and the calculated shape of the concave mirror are shown in Fig. 2.

The obtained concave mirror profile has been used further for the open resonator simulation by using CST STUDIO SUITE (Darmstadt, Germany). These simulations take into account the microwave losses in the aqueous sample and the influence of the coupling slit between the cavity and feeding waveguide. The simulated geometry is shown in Fig. 1 with a 80 μm thick water layer on top of the plain mirror covered by a sapphire plate (1.5 mm), thick enough to withstand the pressure applied under flow conditions (up to 7 bar). The dielectric permittivity of pure water with a temperature close to 100 °C at 42 GHz was set to $\varepsilon' = 47$, $\varepsilon'' = 32$ (loss tangent $\tan\delta = 0.68$) (Andryievski et al., 2015). The resonance modes (TEM$_{00}$, TEM$_{02}$, TEM$_{04}$) of the resonator with the respective $B_1$ distribution along the water layer are shown in Fig. 3.

In the simulations, the slit width between the feeding waveguide and the resonator was optimized to achieve the most effective mw excitation. Since the mw absorbance in the water layer is strong, a slit dimension as large as $1.4 \times 5.6$ mm$^2$ is needed to reach the critical coupling. Therefore, the wave scattering by the coupling slit results in some distortion of the $B_1$ field homogeneity at the flat mirror (Fig. 4a). The simulation predicts a $Q$ factor of the TEM$_{00}$ mode of the resonator of 800 and a maximal conversion factor $c$ of about $3\,\mathrm{G\,W^{-\frac{1}{2}}}$. Uniformity of the RF magnetic field over the sample volume is also an important parameter beside the $B_{1\mathrm{max}}$ value (Mett et al., 2019) that can be evaluated by calculating the mean value of $B_1$. In the Fabry–Pérot resonator the $B_1$ mean value over the sample volume is about $1.55\,\mathrm{G\,W^{-\frac{1}{2}}}$ that is slightly higher in comparison to

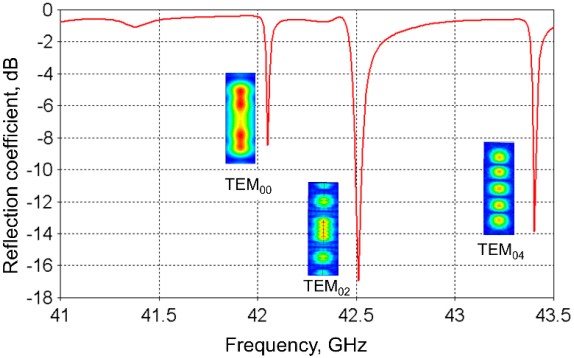

**Figure 3.** Simulated reflection coefficient of the open resonance structure connected to the microwave waveguide. The corresponding resonance modes are identified by calculation of the $B_1$ field distribution at the plain mirror (shown as insets).

the TE$_{013}$ cylindrical cavity ($1.4\,\mathrm{G\,W^{-\frac{1}{2}}}$) (Fig. 4b) used in our MRI DNP experiments previously (Denysenkov et al., 2017). The simulations show that the effectively irradiated area on the plain mirror surface is about $40 \times 12$ mm$^2$, corresponding to 40 μL volume of aqueous solution inside the resonator for an 80 μm thick layer.

## 2.2   MRI DNP setup

MRI detection was performed by a 1.5 T MRI scanner (Magnetom Aera, Siemens, Erlangen, Germany) using a standard 8-channel surface pick-up coil (NORAS, Germany) positioned horizontally in the iso-center of the magnet bore (Fig. 5).

According to the "in-bore" concept, the polarization transfer from the TEMPOL radical electron spins to the water protons as well as the MRI detection are accomplished at ambient temperature in the same 1.5 T magnetic field. This corresponds to an EPR excitation frequency of 42 GHz (Denysenkov et al., 2017).

All MRI images were taken on a meander-shaped phantom made of a 0.4 mm ID and 300 mm long PTFE tubing shown in Fig. 6, in which the hyperpolarized solution from the resonator was transferred via a piece of 0.25 mm ID quartz capillary (Polymicro Technologies, USA). The phantom was placed directly on top of the pick-up coil for optimal sensi-

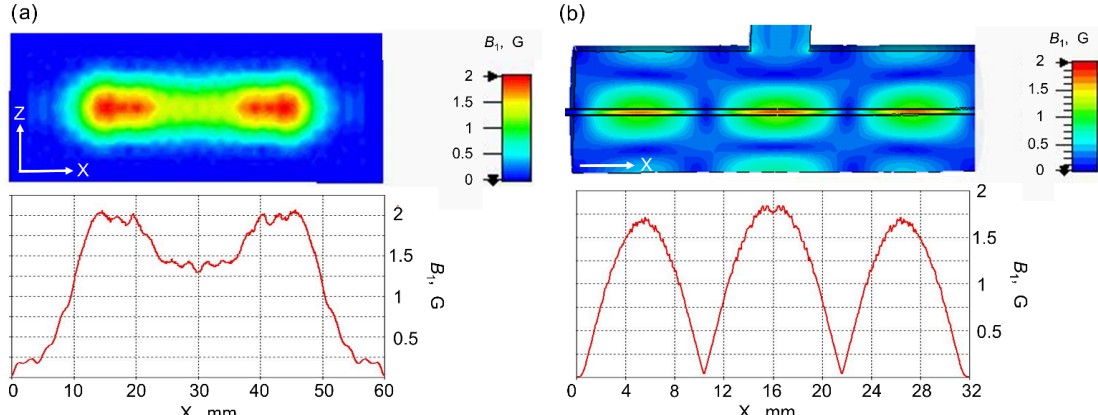

**Figure 4.** Comparison of field distribution of open resonator with cylindrical resonator. **(a)** Calculated $B_1$ field distribution at the plain mirror surface of the Fabry–Pérot resonator in presence of an aqueous layer of 80 µm for the $TEM_{00}$ mode; **(b)** $B_1$ distribution along the 0.5 mm ID quartz capillary with water along the $TE_{013}$ cylindrical cavity. The $B_1$ value is calculated for an input mw power of 0.5 W.

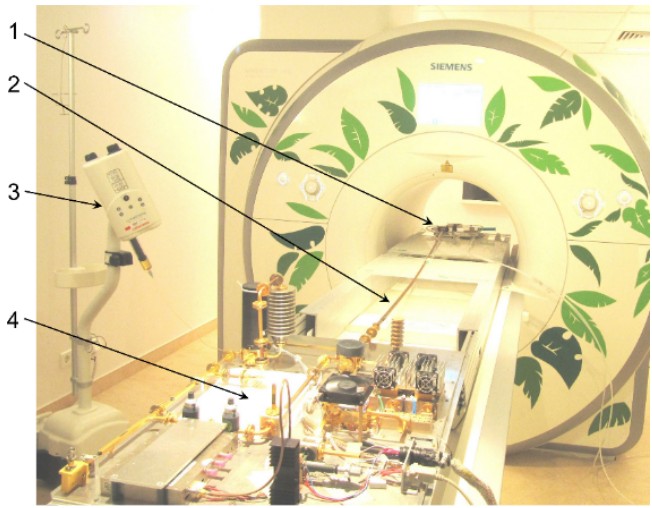

**Figure 5.** In-bore Overhauser DNP-MRI setup. 1 – mw resonator, 2 – waveguide between the mw resonator and the microwave board, 3 – syringe pump, 4 – microwave board.

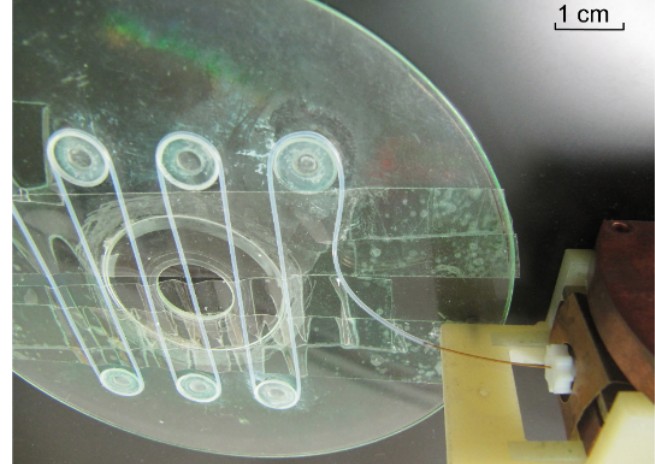

**Figure 6.** Phantom used for the DNP-MRI measurements. The meander-shaped 0.4 mm ID PTFE tubing is mimicking a blood vessel. Inlet of the phantom is connected to the resonator via a piece of 0.25 mm ID quartz capillary (right-bottom corner of the picture).

tivity. For imaging the standard 2D slice-selective gradient echo (GRE) experiment with Cartesian $k$-space filling was used with the number of encoding steps of 144, a repetition time (TR) of 110 ms, an echo time (TE) of 3.8 ms, a pixel bandwidth of 310 Hz, an acquisition matrix of $192 \times 256$ and a field-of-view (FOV) of $90 \times 120$ mm$^2$. The slice thickness was chosen in a range of 3 mm to minimize TE and signal-to-noise losses due to flow induced dephasing. The excitation RF pulse flip angle (FA) was set to 65° to meet the optimal conditions for the best signal contrast and sensitivity. The signal-to-noise-ratio (SNR) of the reference image (without DNP) was improved by acquiring 3 averages per scan resulting in an overall duration of 47 s. The SNR was determined by measuring the maximal signal intensity at the inlet area of the phantom and the standard deviation of the noise outside the phantom.

The nitroxide radical TEMPOL (4-hydroxy-2,2,6,6-tetramethylpiperidine-1-oxyl, Sigma-Aldrich, USA) was used as DNP agent to polarize water protons. The best DNP enhancements were obtained with a concentration of 28 mM TEMPOL, which was optimized previously (Denysenkov et al., 2017). Measurements of DNP enhancements and SNR were done for flow rates of the TEMPOL aqueous solution ranging from 0.5 to 4 mL min$^{-1}$. Such flow rates of the aqueous solution have been accomplished by using a 10 mL syringe driven by an ALADDIN syringe pump (WPI Inc., USA), and injected into the resonator via a 0.8 mm ID PTFE tube.

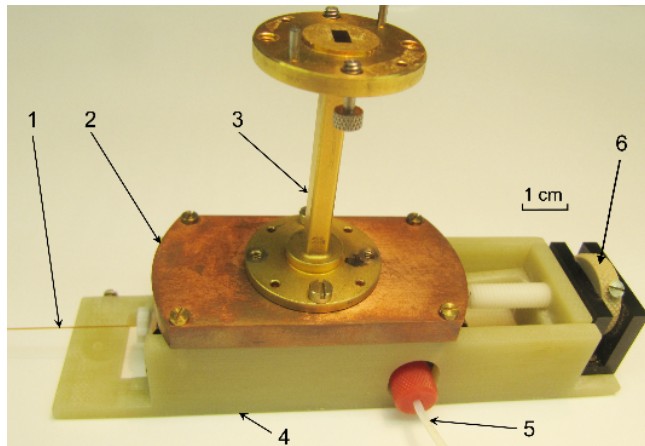

**Figure 7.** The resonator assembly. 1 – quartz capillary from the resonator to phantom, 2 – concave mirror, 3 – WR-22 waveguide, 4 – housing including the plain mirror, 5 – 0.8 mm ID tubing from the syringe filled with the TEMPOL solution, 6 – frequency tune knob.

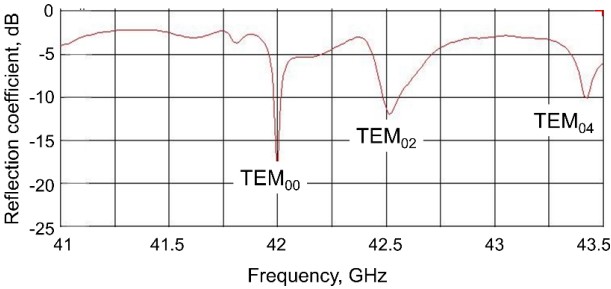

**Figure 8.** Experimentally measured reflection coefficient of the open resonator connected to the microwave waveguide. The water layer is $50\,\mu\mathrm{m}$ thick, the sapphire window is 1.5 mm thick, and the separation between mirrors is $2\lambda$. The different resonance modes are identified by comparison with the CST simulations shown in Fig. 3.

## 3 Experimental Results

### 3.1 Resonator performance

The open resonator (Fig. 7) was home-build in the workshop of the Institute of Physical and Theoretical Chemistry at the Goethe University Frankfurt and tested experimentally for its mw performance with a Rohde & Schwarz ZVA-40 network analyzer (Munich, Germany).

The measured frequency response showed several resonance modes (Fig. 8) which can be identified according to the resonance frequencies calculated previously by CST (compare with Fig. 3).

The $\mathrm{TEM}_{00n}$ mode was chosen for the DNP experiments due to its high homogeneity of the $B_1$ field distribution along the flow axis and the highest average value of $B_1$ over the sample. The longitudinal mode number $n$ defines the number of $\lambda/2$ wavelengths between the two mirrors. It was set

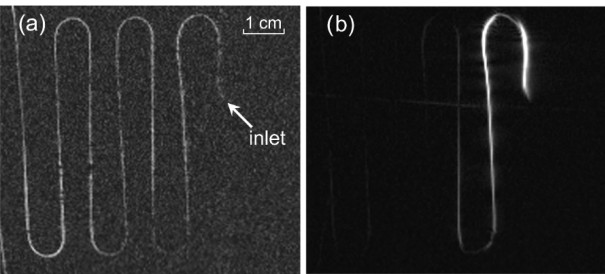

**Figure 9.** MRI image of TEMPOL aqueous solution flowing through the phantom. **(a)** Without microwaves, 3 acquisitions; **(b)** with microwaves, single acquisition. Phantom inlet position is indicated by the arrow. The radical concentration is 28 mM and the flow rate is $2.5\,\mathrm{mL\,min^{-1}}$.

to $n = 4$ for our experiments to reach a critical mw coupling to the resonator and avoid any reflection losses. The measured $Q$ value of this resonance mode is about 700 when the resonator is filled with a $50\,\mu\mathrm{m}$ thick water layer on top of the plain mirror. It resulted in the conversion factor of $1.55\,\mathrm{G\,W^{-\frac{1}{2}}}$ (mean value) over the sample volume, which agreed very well with the value calculated by the CST software. This demonstrates that finite element calculations are as good as measurements in a well-characterized resonator (Rinard and Eaton, 2005).

### 3.2 MRI DNP Results

MRI images of the TEMPOL/water solution flowing through the meander-shaped phantom were taken to investigate the DNP performance with the new setup. Figure 9 shows the images taken with a GRE sequence with the parameters described before. The substrate flows from the right side to the left side. The image of the radical solution without mw irradiation in the resonator is shown in Fig. 9a. In this case, three acquisitions were necessary to get an appropriate SNR. The water signal at the inlet is weaker because the first loop of the tubing moves out of the phantom plane and therefore leaves the MRI slice (see Fig. 6). The image on the right side (Fig. 9b) was obtained as a single acquisition with the microwaves turned on to its maximum power of 10 W at the microwave board. Both pictures are taken with the same flow rate of $2.5\,\mathrm{mL\,min^{-1}}$. The signal intensity under DNP conditions at the inlet is 18-fold larger compared to the control experiment without mw irradiation. As can be seen from the picture the enhancement fades away along the pathway of the flowing liquid and disappears at approximately 22 cm.

MRI images were taken also at different flow rates to quantify the superior performance of the new open resonator (with the $\mathrm{TEM}_{004}$ mode) compared to the cylindrical $\mathrm{TE}_{013}$ cavity. The signal enhancement (ratio of signal with and without mw) at the inlet versus different flow rates for both resonators is shown in Fig. 10. As can be seen the signal enhancement is a factor of 2 larger with the Fabry–Pérot resonator compared

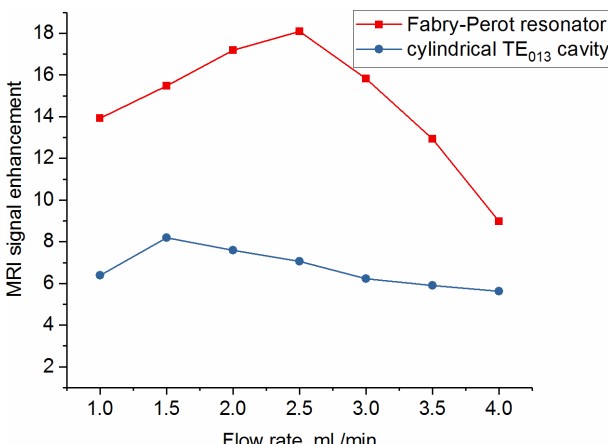

**Figure 10.** DNP MRI signal enhancements as a function of the flow rate. Shown are the results with the new Fabry–Pérot resonator (red) and the cylindrical cavity (blue) at a mw power of 10 W.

to the cylindrical one. The maximum enhancement of 18 is reached for a flow rate of 2.5 mL min$^{-1}$.

## 4   Discussion

The experimental tests showed that the new resonator has a good mw performance only if the thickness of the water layer is reduced to 50 µm with respect to the value of 80 µm used in the CST simulations. This can be explained by imperfections of the real device geometry caused by fabrication steps, as well as by the higher mw losses ($\tan\delta = 1.2$) in the water sample at room temperature (Ellison et al., 1996), which was the experimental temperature at the mw characterization of the device with a network analyser with low amount of mw output power (no heating). In the DNP experiments the temperature of the sample is between 65 and 95 °C due to the heating by the much higher applied mw power of 10 W. The higher temperature of the water solution in the DNP resonator shortens the rotational and translational correlation times improving the maximum achievable signal enhancements. Additionally this increases the flow rate inside the resonator structure slightly due to fluid expansion. After leaving the mw resonator the temperature of the hyperpolarized solution in the outlet capillary can be brought back to any desired temperature by a heat-sink chiller as described before (Denysenkov et al., 2017) to be physiologically compatible with in vivo experiments.

The optimum flow rate for signal enhancement by DNP is a trade-off between the dwell time of the liquid inside the mw resonator and the flow time through the phantom. The first aspect requires low flow rates to achieve maximum polarization build-up, which is on the timescale of the proton spin relaxation time $T_1$. The second aspect requires high flow rates to reduce the hyperpolarization losses due to proton $T_1$ relaxation determined by the flow time through the phantom.

As a consequence of this second point, the highest enhancement is observed at the inlet of the phantom due to the shortest time delay after the DNP polarization process inside the resonator.

The DNP enhancement observed at this point is more than a factor of 2 larger for the new open Fabry–Pérot resonance structure compared to the cylindrical cavity (Fig. 10). The reason is the much larger sample volume inside of the new resonator (25 µL) compared to the cylindrical cavity (6.3 µL), leading to a much longer residence time of the liquid under flow conditions and therefore a larger starting polarization. The achieved $B_1$ field strength with the Fabry–Pérot resonator is only 10 % smaller compared to the cylindrical cavity for the TE$_{013}$ mode. This is due to the fact that higher $Q$ values can be achieved with the new resonance structure by minimizing the electrical field component of the mw at the sample along the full pathway within the resonator. This cannot be fulfilled as well for the higher mode cylindrical cavity. Similarly, the $B_1$ field in the new resonator is much more homogeneous along the pathway of the sample compared to the cylindrical cavity. All these factors lead to an expected larger DNP enhancement of the liquid at the exit of the new resonator.

The hyperpolarization of the liquid relaxes exponentially back to the thermal Boltzmann equilibrium polarization after leaving the DNP resonator with a time constant given by the proton $T_1$ relaxation time. For the 28 mM TEMPOL solution this time is approximately 0.17 s. Depending on the flow characteristics (laminar, bolus or turbulent), this would lead to a decrease in the polarization of $1/e$ after approximately 10–15 cm path length inside of the phantom. The experimentally observed decay is much faster due to the chosen gradient echo pulse sequence with a short TR of 110 ms. This time is much shorter than the travel time of the hyperpolarized liquid within the meander-shaped pathway of the phantom. Therefore the hyperpolarized liquid is excited several times by the RF $\pi/2$ pulse of the gradient echo sequence. Under our experimental conditions (flip angle of 65°), each subsequent pulse reduces the hyperpolarization by more than a factor of 2. This can be avoided by applying fast imaging techniques such as echo planar imaging (EPI or SE-EPI) or/and SPatio-temporal ENcoding (SPEN) sequences that are single RF pulse experiments, instead of the multi-pulse GRE protocol used here. However, these sequences are technically more demanding and have to be optimized with respect to contrast and flow artifacts. The fast polarization decay along the flowing liquid due to the short proton spin relaxation time induced by the paramagnetic radicals could be removed by immobilizing the TEMPOL radical inside the DNP resonator (McCarney and Han, 2008; Gajan et al., 2014). This would not only allow the hyperpolarized liquid pathway inside the blood vessels to be observed for a prolonged time and length after the injection point, but also would avoid the exposure of the object to the radical.

The image of the DNP enhanced (bright) signal along the phantom tubing has some dark regions at the positions where the tubing is bent (see Fig. 9b). This feature can originate from some turbulence of the flow at these positions, leading to intra-voxel dephasing, which depends on the degree of turbulence and the direction of the phase- and frequency-encoding gradients with respect to the flow direction (Westbrook et al., 2011). To avoid these additional complications, we compare here the maximum achievable DNP polarization at the inlet point (approximately 5 cm after the resonator). With the new type of resonator, not only larger enhancements, but also a maximum at larger flow rates, was achieved. This is again due to the larger sample volume inside the resonator. Therefore larger amounts and stronger hyperpolarized liquids can be achieved with the new resonator type.

## 5   Conclusions

The developed open Fabry–Pérot resonator for Overhauser DNP exhibits an enlarged aqueous sample volume of 24 μL inside the mw resonator, a 4-fold increase with respect to a cylindrical $TE_{013}$ cavity at 42 GHz. Additionally, the new resonator has a more homogeneous amplitude of the mw $B_1$ field along the sample pathway inside the resonance structure. The design of the resonator and sample geometry also leads to a reduced $E_1$ mw electrical field component along the sample pathway. This leads to a larger $Q$ value of 700 and thus to a similar efficient $B_1$ field amplitude (1.3 G) at the sample compared to a cylindrical $TE_{013}$ cavity. First MRI DNP experiments with the new resonator demonstrate the possibility of increasing the flow rate of hyperpolarized solutions significantly (4-fold) as well as of reaching higher DNP enhancements (2-fold). The obtained improvement is encouraging and might together with using immobilized TEMPO radicals be a promising approach for in vivo MRI angiography applications on small animals in the future.

**Data availability.** The Supplement contains the following files: (1) Fabry_Perot resonator simulation.cst that was used to optimize geometry of the structure and (2) Fabry_Perot resonator design.dwg that was used to show all the parts and dimensions of the structure. TS2

**Supplement.** The supplement related to this article is available online at: https://doi.org/10.5194/mr-1-1-2020-supplement.

**Author contributions.** AF simulated the resonator structure and wrote Sect. 2 of the manuscript. IK calculated the concave mirror profile. SF optimized the MRI protocol and wrote Sect. 3 of the manuscript. TV supervised the MRI experiments. TFP supervised the DNP project and wrote Sects. 1 and 4 of the manuscript. VD planned the resonator improvements, designed the device, performed the resonator performance tests, and participated in the MRI

experiments. He also wrote the final version of the manuscript. All the authors read and approved the final manuscript.

**Competing interests.** The authors declare that they have no conflict of interest.

**Special issue statement.** This article is part of the special issue "Robert Kaptein Festschrift". It is not associated with a conference.

**Acknowledgements.** We acknowledge additional financial support by the Hessian Center of Biomolecular Magnetic Resonance (BMRZ). TS3

**Financial support.** This research has been supported by the DFG (Deutsche Forschungsgemeinschaft, grant no. 405972957) and by the Russian Ministry of Education and Science (grant no. 0035-2019-0001).

This open-access publication was funded by the Goethe University Frankfurt.

**Review statement.** This paper was edited by Konstantin Ivanov and reviewed by two anonymous referees.

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

## Remarks from the typesetter