# Peer review of "Increased flow rate of hyperpolarized aqueous solution for DNPenhanced MRI achieved by an open Fabry-Pérot type microwave resonator"

_Magnetic Resonance, 2020_

## Referee Comment (RC1) · Anonymous Referee #1 · 15 Sep 2020

mr-2020-20 manuscript "Increased flow rate of hyperpolarized aqueous solution for DNP-enhanced MRI achieved by an open Fabry-Pérot type microwave resonator" Alexey Fedotov, Ilya Kurakin, Sebastian Fischer, Thomas Vogl, Thomas F. Prisner, and Vasyl Denysenkov

General comments:

This is a new and thorough study describing the development and testing of new open Fabry-Perot resonator for applications in DNP/MRI experiments. The research is topical and well justified; the results are of high quality and are nicely presented in the paper. The authors achieved two-fold increase in signal enhancement compared to cylindrical resonator in similar conditions, but more important, they outlined possible directions for further development in continuous flow polarizers for DNP. The research is well planned, it starts from simulations and continues to construction of the resonator and then its testing using phantom system inside 1.5 T MRI scanner.

In my opinion, this paper can be published as is. However, I provide some comments which authors might wish to consider in the final version of the paper.

Specific comments:

1. Thinking about a broad reader in fields of NMR/EPR/DNP, it would be good to explain the choice/influence of such parameters used as TEMPOL concentration (28 mM) and temperature of water in the resonator. Do these parameters impact the generation/relaxation of polarization critically in current setup or not?

2. It is conceivable that heating of water by microwaves lowers its viscosity and makes the flow faster. The comment on pros and cons of this would be welcome, especially since the 'flow rate' is in the title of the paper.

3. Can any strategies of external cooling be foreseen for this type of the resonator?

Technical corrections:

The paper is written very well and clearly. Only a few typos were found:

Line 226: "volume, which agreed very with the value" lacks the word "well"

Lines 316-317: please, change "planed" to "planned", and maybe reformulate "supervised DNP supervision"

---

## Referee Comment (RC2) · Anonymous Referee #2 · 21 Sep 2020

Focus of the manuscript is the design and construction of a microwave structure that allows for improved saturation of EPR transitions in Overhauser DNP applied to MRI investigations at high magnetic field (1.5 T). The development of the hardware is a very nice piece of work; the achieved signal enhancement is quite substantial. The potential in real MRI application, however, is not satisfactorily discussed, whereas the problems of alternative methods got much space in the Introduction; for example, drawbacks of Gd-based contrast agents are discussed in detail, while disadvantages of nitroxide dopants as used here stay unmentioned. The concluding sentence of

the manuscript concerning angiography applications is, therefore, not compelling and must be modified. In addition, part of the text seems to be written only for the small group of microwave specialists, but not for the general magnetic resonance community. As an example, the parameter S11 in Figures 3 and 8 is taken as granted, but should be defined; in Figure 4 and in Conclusion the microwave field B1 is given in the strange unit of A/m. Some modifications in this regard could improve the manuscript considerably. Also, reason for the choice of TEMPOL as radical dopant is missing; no word is lost about the concentration of 28 mM, although it probably affects the proton T1 relaxation, which is a crucial factor for any application of the method.

Please also note the supplement to this comment:
https://mr.copernicus.org/preprints/mr-2020-20/mr-2020-20-RC2-supplement.pdf

**Supplement:**

**Reviewer Report**

**Manuscript Number:** mr-2020-20

**Title:** "Increased flow rate of hyperpolarized aqueous solution for DNP-enhanced MRI achieved by an open Fabry-Pérot type microwave resonator"

**Authors:** Alexey Fedotov, Ilya Kurakin, Sebastian Fischer, Thomas Vogl, Thomas F. Prisner, Vasyl Denysenkov

Focus of the manuscript is the design and construction of a microwave structure that allows for improved saturation of EPR transitions in Overhauser DNP applied to MRI investigations at high magnetic field (1.5 T). The development of the hardware is a very nice piece of work; the achieved signal enhancement is quite substantial. The potential in real MRI application, however, is not satisfactorily discussed, whereas the problems of alternative methods got much space in the Introduction; for example, drawbacks of Gd-based contrast agents are discussed, while disadvantages of nitroxide dopants as used here stay unmentioned. The concluding sentence of the manuscript on angiography applications is, therefore, not compelling and must be modified.

Also, part of the text seems to be written only for the small group of microwave specialists, but not for the general magnetic resonance community. As an example, the parameter $S_{11}$ in Figures 3 and 8 is taken as granted, but should be defined; in Figure 4 and in Conclusion the microwave field $B_1$ is given in the strange unit of A/m. Some modifications could improve the manuscript considerably. Also, reason for the choice of TEMPOL as radical dopant is missing; choosing the concentration of 28 mM stays unsubstantiated, although it probably affects the proton $T_1$ relaxation.

In summary, the manuscript is not yet coherently written and needs modification before it is suited for publication.

There are a few minor points listed below that the authors should also reconsider.

line 34: change to "optically"

line 175, Figure 5: use a photo of better contrast

line 210, Figure 7: define the dark brown parts on the right side of the assembly

line 316: change: "supervised DNP supervision"

---

## Author Comment (AC1) · 7 Oct 2020

Dear Konstantin,

attached you find our detailed answers to all the questions and remarks of both reviewers together with our accordingly revised manuscript (with all changes are highlighted in yellow). We are thankful for the general positive evaluation of both reviewers of our experimental work

5   and on our improved MRI-DNP enhancements achieved under continuous flow conditions and we revised our manuscript according to the specific comments and suggestions of both reviewers, regarding the specificity of our DNP approach using nitroxides and its potential use for medical applications in MRI. We hope very much that with this issues discussed in more detail our manuscript can now be accepted for publication in the Magnetic Resonance journal.

10  With best regards

Thomas (on behalf of all authors)

Comments to Reviewers:

**Reviewer 1:**

This is a new and thorough study describing the development and testing of new open Fabry-Perot resonator for applications in DNP/MRI experiments. The research is topical and well justified; the results are of high quality and are nicely presented in the paper. The authors achieved two-fold increase in signal enhancement compared to cylindrical resonator in similar conditions, but more important, they outlined possible directions for further development in continuous flow polarizers for DNP. The research is well planned, it starts from simulations and continues to construction of the resonator and then its testing using phantom system inside 1.5 T MRI scanner. In my opinion, this paper can be published as is.

We are grateful to this referee for the positive evaluation of our work and the useful comments (see below).

However, I provide some comments which authors might wish to consider in the final version of the paper.

Specific comments: 1. Thinking about a broad reader in fields of NMR/EPR/DNP, it would be good to explain the choice/influence of such parameters used as TEMPOL concentration (28 mM) and temperature of water in the resonator. Do these parameters impact the generation/relaxation of polarization critically in current setup or not?

Many thanks for this comment. The TEMPOL concentration was optimized in our previous study by measuring the DNP enhancement for concentrations in the range between 20 mM and 40 mM. We have added this comment to the manuscript:

> *The best DNP enhancements have been obtained with a concentration of 28 mM TEMPOL as described previously (Denysenkov et al., 2017).*

2. It is conceivable that heating of water by microwaves lowers its viscosity and makes the flow faster. The comment on pros and cons of this would be welcome, especially since the 'flow rate' is in the title of the paper.

We discuss the pros and cons of the microwave heating now at several points in the revised manuscript in more detail.

 3. Can any strategies of external cooling be foreseen for this type of the resonator?

External cooling of the resonator body itself is of course easily possible. However, as described in the manuscript, it would reduce the DNP enhancement because of the rise of the DNP coupling factor with temperature. Therefore, it would only be necessary if overheating inside the resonator occurs, which is not the case for our available microwave power. Otherwise cooling the hyperpolarized liquid after the resonator by a chiller, as mentioned in the manuscript and published by us earlier is the more efficient way.

Technical corrections: The paper is written very well and clearly. Only a few typos were found: Line 226: "volume, which agreed very with the value" lacks the word "well"

Thank you, is corrected in the revised manuscript.

Lines 316-317: please, change "planed" to "planned",

Thank you; is corrected in the revised version of the manuscript.

and maybe reformulate "supervised DNP supervision"

Thank you, is reformulated in the revised manuscript

**Reviewer 2:**

Focus of the manuscript is the design and construction of a microwave structure that allows for improved saturation of EPR transitions in Overhauser DNP applied to MRI investigations at high magnetic field (1.5 T). The development of the hardware is a very nice piece of work; the achieved signal enhancement is quite substantial.

We are grateful to this referee for the positive evaluation of our experimental work and also the critical comments below.

The potential in real MRI application, however, is not satisfactorily discussed, whereas the problems of alternative methods got much space in the Introduction; for example, drawbacks of Gd-based contrast agents are discussed in detail, while disadvantages of nitroxide dopants as used here stay unmentioned.

The reviewer is right. We did not intend to claim that DNP with nitroxide is superior to the other approaches. Therefore, we elaborate in the introduction and also in the discussion of our revised

manuscript in more detail on the advantages and also the limitations of nitroxide spin labels for DNP and MRI applications.

The concluding sentence of the manuscript concerning angiography applications is, therefore, not compelling and must be modified.

See above. We modified our statement more conservatively.

In addition, part of the text seems to be written only for the small group of microwave specialists, but not for the general magnetic resonance community.

As an example, the parameter S11 in Figures 3 and 8 is taken as granted, but should be defined; in Figure 4 and in Conclusion the microwave field B1 is given in the strange unit of A/m. Some modifications in this regard could improve the manuscript considerably.

Many thanks. We changed S11 to reflection coefficient in Fig. 3 and 8 and we give the microwave field now in Gauss in Figure caption 4 and in the text of the revised manuscript.

Also, reason for the choice of TEMPOL as radical dopant is missing; no word is lost about the concentration of 28 mM, although it probably affects the proton T1 relaxation, which is a crucial factor for any application of the method.

The TEMPOL concentration was optimized in our previous study; we added this sentence in the manuscript (see comment to reviewer 1 above). Additionally we included a description of the potential procedure to remove the radical from solution by immobilization.

In summary, the manuscript is not yet coherently written and needs modification before it is suited for publication.

We believe that our revised manuscript is substantially improved by taking the constructive comments of both reviewers into consideration and can now be accepted for publication in *Magnetic Resonance*.

There are a few minor points listed below that the authors should also reconsider.

line 34: change to "optically"

Thank you, has been changed.

line 175, Figure 5: use a photo of better contrast line 210,

Thank you, has been changed.

95 Figure 7: define the dark brown parts on the right side of the assembly

A new position 6 is added to Figure 7 showing the frequency tune knob and explained in the corresponding caption.
line 316: change: "supervised DNP supervision"

Thanks, has been changed.

[revised manuscript text omitted]